# Antifeedant Mechanism of *Dodonaea viscosa* Saponin A Isolated from the Seeds of *Dodonaea viscosa*

**DOI:** 10.3390/molecules27144464

**Published:** 2022-07-12

**Authors:** Hang Yu, Jinliang Li, Guoxing Wu, Qingbo Tang, Xiuan Duan, Quanjun Liu, Mingxian Lan, Yuhan Zhao, Xiaojiang Hao, Xiaoping Qin, Xiao Ding

**Affiliations:** 1State Key Laboratory for Conservation and Utilization of Bio-Resources in Yunnan, Yunnan Agricultural University, Kunming 650100, China; yuhang9604@163.com (H.Y.); lijingliang2021@163.com (J.L.); wgx1@163.cm (G.W.); lqj2my@163.com (Q.L.); lmx4164@163.com (M.L.); 2State Key Laboratory of Phytochemistry and Plant Resources in West China, Kunming Institute of Botany, Chinese Academy of Sciences, Kunming 650201, China; zhaoyuhan@mail.kib.ac.cn (Y.Z.); haoxj@mail.kib.ac.cn (X.H.); 3Department of Entomology, Henan Agricultural University, Zhengzhou 450002, China; qbtang@henau.edu.cn; 4Agro-Environmental Monitoring Center of Baoshan City, Green Development Center of Baoshan City, Baoshan 678000, China; dxa6666@163.com; 5Graduate Student Department, Yunnan Agricultural University, Kunming 650100, China

**Keywords:** antifeedant, *Dodonaea viscosa*, *Spodoptera litura*, taste sensillum, GABA, detoxification enzyme

## Abstract

*Dodonaea viscosa* is a medicinal plant which has been used to treat various diseases in humans. However, the anti-insect activity of extracts from *D. viscosa* has not been evaluated. Here, we found that the total saponins from *D. viscosa* (TSDV) had strong antifeedant and growth inhibition activities against 4th-instar larvae of *Spodoptera litura*. The median antifeeding concentration (AFC_50_) value of TSDV on larvae was 1621.81 μg/mL. TSDV affected the detoxification enzyme system of the larvae and also exerted antifeedant activity possibly through targeting the γ-aminobutyric acid (GABA) system. The AFC_50_ concentration, the carboxylesterase activity, glutathione S-transferases activity, and cytochrome P450 content increased to 258%, 205%, and 215%, respectively, and likewise the glutamate decarboxylase activity and GABA content to 195% and 230%, respectively, in larvae which fed on TSDV. However, *D. viscosa* saponin A (DVSA) showed better antifeedant activity and growth inhibition activity in larvae, compared to TSDV. DVSA also exerted their antifeedant activity possibly through targeting the GABA system and subsequently affected the detoxification enzyme system. Further, DVSA directly affected the medial sensillum and the lateral sensillum of the 4th-instar larvae. Stimulation of *Spodoptera litura*. with DVSA elicited clear, consistent, and robust excitatory responses in a single taste cell.

## 1. Introduction

*Dodonaea viscosa* (Sapindales: Sapindaceae) is a shrub which forms part of the more than 60 species of *Dodonaea* in the world, which are distributed in tropical and subtropical regions, especially in Oceania [1]. In China, *D. viscosa* is the only recorded species that can be found in Southern Fujian, Taiwan, Guangdong, Hainan, Guangxi, Sichuan and Yunnan, etc.

*D. viscosa* is widely used in China; its seed oil is used to make soap or fuel; its leaves are crushed to treat burns and pharyngitis; roots are used to kill insects and poison fish; and the whole strain can be used to treat rheumatism [2]. In recent years, research on this plant has mostly been on its physiological and ecological applications [3,4,5]. At present, there are a few studies which have been conducted on the chemical constituents of *D. viscosa* and their pharmacological activities. Flavonoids [6] and triterpenoids [7,8,9] have been isolated from *D. viscosa*, and have been proven to show anti-inflammatory [10,11], anti-ulcer, anti-spasm [12], anti-viral activities [13], as well as inhibitory effects on ATP citrate lyase [14]. Crude extracts of *D. viscosa* showed good insecticidal activity against a variety of lepidopteran insects, antifeedant activity against *Plutella xylostella*, *Pieris brassicae*, *Helicoverpa armigera*, *Mythimna separata*, and toxic activity against adult *Sitophilus oryzae* and molluscicidal activity. However, only one active compound, *dodoneaviscoside A*, has been identified and successively isolated from the seeds of *D. viscosa* [15].

*Spodoptera litura* (Fab.) (Noctuidae: Lepidoptera) is one of the most pervasive pests, infesting more than 300 crop species worldwide [16]. The larvae can seriously damage soybean, cotton, tobacco, and cruciferous vegetables and other important economic crops [17], and losses due to feeding ranging from 26 to 100% are possible in the field [18]. Due to the high reproductive capacity and migration over large distances in the adult stage, its populations can expand rapidly and move across fields quickly. At present, chemical pesticides are still the main effective method to control *S. litura* in many countries, but some chemical pesticides have had detrimental effects. For example, organochlorine pesticides are banned in China because of their long residual period, difficult decomposition and environmental pollution. Therefore, studies on naturally-derived or plant-originated antifeedants have become important sources of pesticides [19,20,21,22]. The primary mechanisms of antifeedants are through their effects on the taste sensillum, which include three aspects. One is the stimulation of a special sensillum, and the other is the changes caused to the activity of sensillum that sense other compounds, or a combination of the two [23]. Taste sensillum plays an important role in the feeding behavior of insect larvae [24,25] and in lepidopteran larvae for instance, which are located in the medial and lateral sensillum of the maxillary gland [26]. Although most antifeedants act directly on the taste system of insects, some other possible mechanisms have been reported as well. For example, some antifeedants act through antagonizing the action of the γ-aminobutyric acid (GABA) on insect neurons to induce feeding deterrence. For example, the P450 content in *Manduca sexta* was reported to have increased after feeding on nicotine, an antifeedant, and the induced P450-mediated detoxification activity permitted increased consumption of a toxic plant compound [27].

Although several studies have reported on the insecticidal activities of crude extracts of *D. viscosa*, few studies have focused on the insecticidal active compound and its insecticidal mechanism. Therefore, this study identified the major insecticidal active compound of *D. viscosa* and its insecticidal mechanism by phytochemistry, electrophysiology, enzyme activity, and bioassays.

## 2. Results

### 2.1. Antifeedant Activity of TSDV

To evaluate the antifeedant activity of TSDV on *S. litura*, the leaf disc method was used. Results on the feeding area and inhibition rates are shown in Table 1. The feeding area of *Spodoptera litura,* which were exposed to the TSDV diet at concentration of 500–5000 μg/mL, was significantly smaller than those exposed to the control diet. The inhibition rate of 3875 μg/mL TSDV and 500 μg/mL Azadirachtin did not show significant difference (Tukey post hoc test, *p* < 0.05). These results revealed that TSDV has good antifeeding activity. The regression equation of the antifeedant activity was y = 1.4557x + 0.3573 (R^2^ = 0.9641). The AFC_10_, AFC_30_, and AFC_50_ values of TSDV on 4th-instar larvae were 204.17, 676.08, and 1621.81 μg/mL, respectively.

### 2.2. Effects of TSDV on Life-History Traits

To further evaluate the toxic effect of TSDV, its effect on the life-history traits of *S. litura* were recorded. The results showed that the duration of the 4th–6th instar stage of *S. litura* larvae, which were exposed to the TSDV diet at the concentrations of AFC_30_ and AFC_50,_ was significantly longer than those exposed to the control diet (CK) (Table 2). The durations of the pupal stage of females, which were exposed to the TSDV diet and control diet, did not show significant differences. The duration of the male pupal stage, female adult stage, and male adult stage of larvae exposed to the TSDV diet at the concentrations of AFC_30_ and AFC_50_ were significantly shorter than those exposed to the control diet. The pupation rate and emergence rate of larvae, which were exposed to the TSDV diet at the concentrations of AFC_30_ and AFC_50,_ were significantly lower than those exposed to the control diet (Table 3). These results revealed that TSDV interfered with the normal development and metamorphosis established at different stages of *S. litura* in a dose-dependent manner.

### 2.3. Time-Course of the Change in Feeding Response to TSDV

*S. litura* exposed to the control and TSDV diet were observed every 6 has shown in Figure 1A,B. The food intake of *S. litura* offered the control diet during the 0–6 h period was significantly less than those who were offered the control diet during the 6–12 h period. However, for the *S. litura* offered the TSDV diet, the food intake during 12–18 h was significantly less than during the 18–24 h period (Tukey post hoc test, *p* < 0.05). Thus, *S. litura* fed the TSDV diet displayed a 12h time lag in their first significant increase in diet consumption.

### 2.4. Time-Course of Detoxification Enzyme Induction by TSDV

Detoxification enzymes play important roles in the metabolism of xenobiotics in insects. The CarE activity, GST-s activity, and the levels of P450 content in treated *S. litura* were measured. The CarE activity of larvae exposed to AFC_50_ TSDV diet increased significantly in 24–42 h, compared to the control diet (Figure 1C–E). At 30 h, the CarE activity increased to 258% (Figure 1C), and the GST-s activity of those exposed to TSDV increased significantly in 24–36 h, compared to the control diet (Tukey post hoc test, *p* < 0.05). At 24 h, the GST-s activity increased to 205% (Figure 1D). Similar results were obtained for the P450 content; the P450 content increased to 215% (Figure 1E). These results indicated that TSDV activated the detoxification enzyme system in *S. litura*.

### 2.5. Effect of Inhibition of Detoxification Enzymes on the Feeding Response to TSDV

TPP, DEM, and PBO are inhibitors of CarE, GST-s, and P450, respectively. To further determine the relationship between the detoxification enzyme system and antifeedant effect, the feeding response by larvae to inhibitors were determined. Pre-treatment with TPP, DEM, and PBO caused dramatic increases in antifeedant activity in *S. litura* which fed on the TSDV diet (Figure 1F,G). We ascertained whether the inhibition of detoxification enzymes would reduce the tolerance of larvae to the TSDV diet. The TPP, DEM, and PBO treatment, in the absence of TSDV, did not influence total feeding time. Among the larvae pre-exposed to the TSDV diet, those treated with TPP, DEM, and PBO spent significantly less time than those without TPP, DEM, and PBO. However, the total feeding times of the larvae fed on the TSDV diet and TPP, DEM, and PBO were still significantly longer than that fed on the control diet and TSDV diet only (Tukey post hoc test, *p* < 0.05). These results suggested that inhibition of CarE, GST-s, and P450 partially increased the antifeedant activity of TSDV. This further indicated that the activation of the detoxification enzyme system resulted in the decline of the antifeedant effect.

### 2.6. Time-Course of GAD Activity and GABA Content Induction by TSDV

GAD can catalyze the decarboxylation of glutamic acid to form GABA, which is an important marker to show the excitement of the nervous system. GABA receptors are important biochemical sites for insecticides action. Results showed that the GAD activity in *S. litura* exposed to the AFC_50_ TSDV diet increased significantly in 3–25 min, compared to the control diet (Figure 2A,B). At 5 min, the GAD activity increased to 195%. The levels of GABA content in larvae which were exposed to AFC_50_ TSDV significantly increased in 5–55 min (Tukey post hoc test, *p* < 0.05). Treatment with TSDV for 45 min increased the GABA content to 230%. These data showed that TSDV obviously increased GABA content.

### 2.7. Isolation of Antifeedant Active Compound

To identify the active compound of TSDV, we used RP-18, MCI, and HPLC to isolate the major active compound from the seeds of *D. viscosa*. The active compound identified was *D. viscosa* saponin A (DVSA) (Figure 3).

### 2.8. Evaluation of the Antifeedant Activities of DVSA

To evaluate whether DVSA was the major active compound responsible for the antifeedant activity of TSDV, the feeding area and inhibition rates were recorded and the results are shown in Table 4. The regression equation of the antifeedant activity was y = 1.0094x + 2.5091 (R^2^ = 0.9721). The AFC_10_, AFC_30_, and AFC_50_ values of DVSA on 4th-instar larvae were 18.2, 92.0, and 288.4 μg/mL, respectively.

Then the effects of DVSA on the life-history traits of *S. litura* were further evaluated. Similar life-history traits were observed in larvae that fed on the DVSA diet compared to the TSDV diet (Table 5 and Table 6). DVSA also interfered with the normal development and metamorphosis established at different stages of *S. litura* in a dose-dependent manner. 

The feeding response against DVSA was then recorded. *S. litura* which fed on the DVSA diet displayed a 12-h time lag in their first significant increase in diet consumption, compared to those that fed on the control diet (Figure 4A,B). This phenotype was also observed in *S. litura* that fed on the TSDV diet. All these results suggested that DVSA might be the active compound of TSDV. 

### 2.9. Time-Course of Detoxification Enzyme Induction by DVSA

To evaluate the detoxification enzyme induction by DVSA, the CarE activity, GST-s activity, and the levels of P450 content of *S. litura* were measured. Compared to the control diet, the DVSA diet increased CarE activity, GST-s activity, and the P450 content level to 228%, 189%, and 247%, respectively (Figure 4C–E). These results indicated that DVSA also activated the detoxification enzyme system in *S. litura*.

### 2.10. Effect of the Inhibition of Detoxification Enzymes on the Feeding Response to DVSA

Pre-treatment with TPP, DEM, and PBO to inhibit detoxification enzymes in larvae caused a similar antifeedant activity in the DVSA diet compared to the TSDV diet (Figure 4F,G). The results suggested that the activation of detoxification enzyme system in larvae which fed on the DVSA diet resulted in the decline of the antifeedant effect, which was observed for the TSDV diet.

### 2.11. Time-Course of GAD Activity and GABA Content Induction by DVSA 

We also examined the GAD activity and GABA content induction by the DVSA diet. The dynamic changes in GAD activity and GABA content in *S. litura* exposed to the AFC_50_DVSA diet was similar to the AFC_50_TSDV diet (Figure 5A,B). At 5 min, the GAD activity increased to 235%, compared to the control diet, whereas at 45 min, the GABA content increased to 220%. These data showed that DVSA obviously increased GABA content.

### 2.12. Effects of DVSA on the Taste Sensillum of S. litura

As the mechanism of antifeedant reaction is believed to be the effect on taste sensillum, we further investigated whether DVSA acted directly on the taste cells of the insects. Cells in the lateral sensillum and medial sensillum responded to DVSA in *S*. *litura* (Figure 6A,E). The medial sensillum was slightly more sensitive than the lateral sensillum, and with a slightly higher firing rate over the whole concentration range. Our results, together with that obtained from a previous study on the effect of diterpenoids on insect feeding behavior, provide more evidence that natural products could directly act on the taste cells of insects.

Sinigrin is a common deterrent compound. To verify whether the target of DVSA were neurons, the sensory responsiveness of *S*. *litura* fed with DVSA, sinigrin, and a mixture of DVSA and sinigrin were measured. The isopotential map showed that the DVSA-sensitive cell fired at the same rate as the sinigrin-sensitive cell (Figure 6B). It indicated that the electrophysiological responses of the same cell in the taste sensillum were induced by DVSA and sinigrin. DVSA had no significant inhibitory effect on sinigrin-induced nerve impulses (Figure 6F). These results suggested that DVSA and sinigrin target the same cell.

Inositol is a common phagostimulants for medial sensillum. To verify whether DVSA can inhibit the activity of an inositol-sensitive cell, the sensory responsiveness of *S*. *litura* fed with DVSA, inositol, and a mixture of DVSA and inositol were measured. The isopotential map showed that feeding on a mixture with 1 mM DVSA caused the inositol-sensitive cell to fire at a similar rate to that with inositol alone (Figure 6C). DVSA inhibited the activity of the inositol-sensitive cell (Figure 6G). These results showed that DVSA suppressed the response of inositol, indicating that DVSA inhibited the activity of the inositol-sensitive cell.

Sucrose is a common phagostimulants for lateral sensillum. Similar results were observed for sucrose-sensitive cells (Figure 6D,H). DVSA inhibited the activity of the sucrose-sensitive cell.

Altogether these results indicated that DVSA was the major active compound responsible for the antifeedant activity of TSDV.

## 3. Discussion

The study of botanical antifeedants have been reported as an important method of pest control. In the present study, we investigated the antifeedant activity and mechanism of *D. viscosa*. Azadirachtin is one of the compounds with the best antifeedant effect. We studied the antifeedant activity of DVSA and found that the antifeedant activity of 500 μg/mL Azadirachtin is equivalent to that of 800 μg/mL DVSA, which shows that DVSA has strong antifeedant activity.

In this study, we observed that DVSA directly affected the taste cells of the medial and lateral sensillum. Messchendorp et.al. found that the relationship between sensory input and antifeedant supporting the hypothesis that the medial deterrent cell directly causes the antifeedant in *Pieris brassicae* [28]. GABA and related aminobutyric acids are known to stimulate feeding and evoke taste cell responses among herbivorous insects of various taxa, such as Orthoptera, Homoptera, Coleoptera, and Lepidoptera [29]. Mitchell found that an isoquinoline alkaloid induced antifeedant activity by affecting the GABA system [30], which was consistent with our results.

We also found that Carboxylesterase and cytochrome P450 were the main targets of the decline of antifeedant effect in the later stage. These results were consistent with those obtained from studies on the effect of nicotine on the feeding response of tobacco moth and poplar secondary metabolites antifeedant activity on *Lymantria dispar*; the CarE activity and GST-s activity in *L. dispar* increased after feeding on the antifeedant poplar secondary metabolites [31,32].

## 4. Materials and Methods

### 4.1. General Experimental Procedures 

Azadirachtin (purity, ≥81%) was purchased from Yunnan Zhongke Biological Industry Co., Ltd. (Kunming, Yunnan, China). The Insect GABA enzyme-linked immunosorbent assay (ELISA) kit, insect glutamate decarboxylase (GAD) ELISA kit, insect CarE ELISA kit, insect GST-s ELISA kit, and insect cytochrome P450 ELISA kit were purchased from Jianglai Biological Co., Ltd. (Shanghai, China). HPLC-grade acetonitrile (J.T. Baker, Phillipsburg, NJ, USA) and ultra-pure water prepared from a Milli-Q purification system (Millipore, MA, USA) were used for semi-preparative HPLC analysis. 

Electrospray ionization (ESI) were recorded on Aglient 1290 UPLC/6540, and the 1D and 2D NMR spectra were measured on the Bruker 500 MHz spectrometer, with TMS as the internal standard. RP-18 column (50 μm, YMC Co., Ltd., Kyoto, Japan) gel, MCI gel (75–150 μm, Sci-Bio Chem Co., Ltd., Chengdu, China), and Sephadex LH-20 (40–70 μm, Amersham Pharmacia Biotech AB, Uppsala, Sweden), were used for column chromatography. Semi-preparative HPLC was performed on an YMC Luna C18 (5 μm; 10 × 250 mm) reversed-phase column.

### 4.2. Plant Material

Seeds of *D. viscosa* were purchased from Yunnan Ecological Technology Co., Ltd. (Kunming, Yunnan, China). Selected seeds were dried at room temperature and formed into powder using a laboratory mill. A voucher specimen (No. 1906023) was deposited at the State Key Laboratory of Phytochemistry and Plant Resource in West China, Kunming Institute of Botany, Chinese Academy of Sciences (CAS).

### 4.3. Insects

*Spodoptera litura* used in this study were obtained from Yunnan Agricultural University, Kunming, Yunnan, China, and were cultured on cabbage leaves at 25 °C.

### 4.4. Assays for Nonselective Antifeedant Activity

The non-selective antifeedant activity of *D. viscosa* extract against 4th-instar larvae of *Spodoptera litura* was determined using the leaf disc method [33]. The punch was used to make the leaf into a leaf disc with a diameter of 1.5 cm. Five concentration gradients of the extract (5000, 2500, 1250, 625, and 312.5 μg/mL) were prepared and used for treatment. Treatment with distilled water served as the control. To each leaf disc, 50 µL of the different concentrations of compound was dripped onto the leaf surface. After 24 h and 48 h of culture, the leaf area of each tested insect was measured with checkerboard paper. The regression equation of the antifeedant activity was obtained by linear regression. AFC_10_, AFC_30,_ and AFC_50_ was calculated. AFC_10_, AFC_30,_ and AFC_50_ are the concentrations when the inhibition rate is 10%, 30%, and 50%, respectively. The experiment was conducted three times, and all treatments were performed with ten samples.

The experiment was completed in June 2020. Starvation began at 8:00 a.m. and ended at 12:00 a.m. Then, the larvae and leaf discs were added to the feeding room, and the leaf discs were replaced every 6 h.

### 4.5. Detoxification Enzymes Assays

The 4th-instar *S. litura* larvae was exposed to the control, AFC_50,_ TSDV, and AFC_50,_ or DVSA diets. The experiment was completed in June 2020. The insects began to starve for 4 h at 8:00 a.m., and at 12:00 p.m., the insects and leaf discs were added to the feeding room. The timing began after the tested insects began to eat the leaf discs. After 6, 12, 18, 24, 30, 36, 42, and 48 h, the tested insects were grinded and centrifuged to obtain the crude enzyme solution and tested.

The insect cytochrome P450 ELISA kit (Jianglai Biological Co., Ltd., Shanghai, China), insect CarE ELISA kit (Jianglai Biological Co., Ltd., Shanghai, China), and insect GST-s ELISA kit (Jianglai Biological Co., Ltd., Shanghai, China) were used for the measurement of cytochrome P450 content, CarE activity, and GST-s activity, respectively, following the manufacturer’s instructions. The experiment was conducted three times, and all treatments were performed with ten samples.

### 4.6. Effect of the Inhibition of Detoxification Enzymes Activities on the Feeding Response in Larvae to D. viscosa

The experiment was completed in July 2020. The insects began to starve for 4 h at 8:00 a.m., and at 12:00 p.m., the insects and leaf discs were added to the feeding room. We measured detoxification enzymes activity in larvae against TSDV (DVSA) after: (i) pre-exposure to control diet for 24 h, (ii) pre-exposure to the TSDV (DVSA) diet for 24 h, or (iii) pre-exposure to the TSDV (DVSA) diet for 22 h, and then a diet containing TSDV (DVSA) plus 300 μg/mL inhibition for the next 2 h.

A group of larvae were pre-exposed to the control diet for 24 h and then exposed to (a) the same diet, (b) the inhibition diet, or (c) the TSDV (DVSA) diet. For the inhibition treatments, we added inhibitory agents to the diets during the last 2 h of the pre-exposure period. We pre-exposed other larvae to the TSDV (DVSA) diet for 24 h and then exposed them to (d) the same diet, or (e) the same inhibition diet. To determine whether prior ingestion of inhibition diet inhibited TSDV (DVSA) consumption, we compared treatments (d) and (e). The experiment was conducted three times, and all treatments were performed with ten samples.

### 4.7. Levels of GABA Content and GAD Activity Assays

The insect GABA ELISA kit (Jianglai Biological Co., Ltd., Shanghai, China) and GAD ELISA kit (Jianglai Biological Co., Ltd., Shanghai, China) were used for the measurement of GABA content and GAD activity, respectively, following the manufacturer’s instructions. The experiment was completed in July 2020. The insects began to starve for 4 h at 8:00 a.m., and at 12:00 p.m., the insects and leaf discs were added to the feeding room. The timing began after the tested insects began to eat the leaf discs. After 1, 2, 3, 4, 5, 15, 25, 35, 45, 55, 65, and 120 min, the tested insects were grinded and centrifuged to obtain the crude enzyme solution and tested. Briefly, the samples were incubated in microtiter wells with appropriate dilution solutions, according to the specific protocol of the kits. Then, biotinylated antibody, HRP-streptavidin, substrate reagent, and stop solution were sequentially added to the wells. Finally, absorbance was read at a wavelength of 450 nm immediately. The experiment was conducted three times, and all treatments were performed with ten samples.

### 4.8. Effects of DVSA on the Taste Sensillum of S. litura

Each chemical (*D. viscosa* saponin A, sucrose, inositol, and sinigrin) was presented in a series of concentrations from 0.1 mM to 10 mM in KCl as an electrolyte. In addition to the single chemicals, we also tested binary mixtures of *D. viscosa* saponin A (DVSA) with sucrose, inositol, or sinigrin. The concentrations of each component in the mixtures were 1 mM.

The single sensillum recording (SSR) method was used to determine the taste electrophysiology of larvae. The experiment was completed in August 2020. Starvation began at 8:00 a.m. and ended at 10:00 a.m. Then, the heads of the larvae were cut off from the first and second thoracic segments of their bodies. One end of the silver wire was bent into a spoon shape, and the head was gently inserted into the incision of the insect’s chest, so that the larvae’s sensillum extended outwards. The wire was connected to a preamplifier with a copper mini-connector. A glass capillary filled with the test compound, into which a silver wire was inserted, was placed in contact with the sensilla. The stimulation solution induced the taste cells in the sensillum to produce action potential. Electrophysiological responses were quantified by counting the number of spikes in the first second after the start of stimulation. The interval between two successive stimulations was at least 3 min to avoid adaptation of the tested sensilla. Before each stimulation, a piece of filter paper was used to absorb the solution from the tip of the glass capillary containing the stimulus solution to avoid an increase in concentration due to evaporation of water from the capillary tip. The temperature during recording ranged from 22 to 25 °C. The electrophysiological signals were recorded by SAPID Tools software version 16.0, and analyzed using Autospike software version 3.7. The experiment was conducted three times, and all treatments were performed with ten samples.

### 4.9. Isolation of DVSA

The extract of the seeds of *D. viscosa* (5 kg) was obtained by extraction with ethanol three times at room temperature and the solvent was evaporated in vacuo. The extract solution was suspended in H_2_O (0.5 L) and subjected to gradual extraction with petroleum ether, ethyl acetate, and *n*-BuOH (3 × 1.5 L). The *n*-BuOH extract was concentrated at a reduced pressure to obtain TSDV. The TSDV (50.5 g) was chromatographed on an RP-18 column. Elution with water-methanol (97:3–0:100) was done to yield six fractions (1–6). Fraction 6 (9.6 g) was separated using a Sephadex LH-20 column eluted with CH_3_OH. Six subfractions (6A–6E) were collected. Fraction 6C (6.2 g) was applied to an MCI gel column (CH_3_OH-H_2_O from 1:9 to 8:2) to yield seven fractions (6C1–6C7). Fr. 6C5 (4.8 g) was purified by a semi-preparative C18 HPLC column with H_2_O-ACN-MeOH (10:1:9) to obtain DVSA (1.2 g).

DVSA was a white amorphous powder and its molecular formula C_57_H_90_O_24_ was determined by the protonated [M − H]^−^ ion peak at *m*/*z* 1157.5745 (calcd 1157.5743) in the HRESI-MS.

White powder, ${\left[\alpha \right]}_{D}^{25}$–34.8 (c 0.20, MeOH), ESI-MS *m*/*z*: 1157.6 [M − H]^−^; ^1^H NMR (CD_3_OD, 125 MHz) *δ*_H_: 1.67 (1H, m, H-1), 1.03 (1H, d, *J* = 8.8 Hz, H-1), 1.76 (1H, m, H-2), 1.45 (1H, s, H-2), 3.22 (1H, m, H-3), 0.80 (1H, s, H-5), 1.45 (2H, m, H-6), 1.77 (2H, m, H-7), 1.62 (1H, s, H-9), 1.97 (2H, m, H-11), 5.48 (1H, s, H-12), 3.77 (1H, m, H-15), 3.86 (1H, m, H-16), 2.68 (1H, m, H-18), 2.62 (1H, m, H-19), 1.21 (2H, m, H-19) 5.89 (1H, d, *J* = 8.0 Hz, H-21), 5.61 (1H, s, H-22), 1.11 (3H, s, H-23), 0.90 (3H, brs, H-24), 1.01 (3H, brs, H-25), 1.03 (3H, brs, H-26), 1.42 (3H, brs, H-27), 3.32 (1H, s, H-28), 3.04 (1H, s, H-28), 0.85 (3H, s, H-29), 1.09 (3H, s, H-30), 3.13 (1H, m, H_21_-3), 1.32 (3H, d, *J* = 4.4 Hz, H_21_-4), 1.48 (3H, s, H_21_-5), 2.41 (1H, q, *J* = 5.6 Hz, H_22_-2), 2.00 (1H, m, H_22_-3), 1.45 (1H, m, H_22_-3), 0.97 (3H, t, *J* = 6.0 Hz, H_22_-4), 1.19 (3H, d, *J* = 5.6 Hz, H_22_-5), 4.53 (1H, m, H-1′), 3.77 (1H, s, H-2′), 3.77 (1H, s, H-3′), 3.51 (1H, s, H-4′), 3.84 (1H, s, H-5′), 4.69 (1H, m, H-1″), 3.58 (1H, s, H-2″), 3.88 (1H, s, H-3″), 3.49 (1H, s, H-4″), 4.13 (1H, s, H-5″), 3.80 (1H, m, H-6″), 3.68 (1H, m, H-6″), 5.28 (1H, brs, H-1‴), 4.13 (1H, brs, H-2‴), 3.86 (1H, s, H-3‴), 4.13 (1H, brs, H-4‴), 3.80 (1H, m, H-5‴), 3.68 (1H, m, H-5‴); ^13^C NMR (CD_3_OD, 500 MHz) *δ*_C_: 40.2 (t, C-1), 27.0 (t, C-2), 91.9 (d, C-3), 40.4 (s, C-4), 56.7 (d, C-5), 19.2 (t, C-6), 37.2 (t, C-7), 42.3 (s, C-8), 48.3 (d, C-9), 38.0 (s, C-10), 24.6 (t, C-11), 126.9 (d, C-12), 143.4 (s, C-13), 48.0 (s, C-14), 68.5 (d, C-15), 73.89 (d, C-16), 48.5 (s, C-17), 41.6 (d, C-18), 47.6 (t, C-19), 36.6 (s, C-20), 81.7 (d, C-21), 73.8 (d, C-22), 28.4 (q, C-23), 16.9 (q, C-24), 16.0 (q, C-25), 17.6 (q, C-26), 21.0 (q, C-27), 63.6 (t, C-28), 29.5 (q, C-29), 19.8 (q, C-30), 171.2 (s, C_21_-1), 60.7 (s, C_21_-2), 61.5 (d, C_21_-3), 13.8 (q, C_21_-4), 19.0 (q, C_21_-5), 178.1 (s, C_22_-1), 42.8 (d, C_22_-2), 27.5 (t, C_22_-3), 12.1 (q, C_22_-4), 16.8 (q, C_22_-5), 104.5 (d, C1′), 78.2 (d, C2′), 86.9 (s, C3′), 70.2 (d, C4′), 76.9 (d, C5′), 171.2 (s, C6′), 104.5 (d, C1″), 73.6 (d, C2″), 77.9 (d, C3″), 77.0 (d, C4″), 77.9 (s, C5″), 63.8 (t, C6″), 110.7 (d, 1‴), 83.2 (d, 2‴), 78.9 (s, 3‴), 85.4 (d, 4‴), 62.5 (t, 5‴).

### 4.10. Statistical Analysis

AFC_10_, AFC_30_, and AFC_50_ values were calculated using GraphPad Prism v7.0.0 and statistical analysis was carried out in SPSS 26. The results presented in the study are given as means ± standard error (SE) from three independent experiments including 10 repeats for each experiment. The one-way analysis of variance (ANOVA) with comparisons of means using Tukey’s honestly significant difference (HSD) test were used to compare inhibition rate, life-history traits, activity of detoxification enzyme, activity of GABA, and response frequency (*p* < 0.05).

## 5. Conclusions

In this study, we found that TSDV had strong antifeedant and growth inhibition activities against *S. litura*, and studied its potential antifeedant mechanism. The results showed that TSDV mainly acted on the detoxification enzyme system and GABA. The active compound in TSDV was isolated and was identified to be DVSA. It was verified that DVSA had similar growth inhibition activity and stronger antifeedant activity on 4th-instar larvae of *S. litura*. Some typical antifeedant symptoms were found from symptomatology, which led to the study of the antifeedant mechanism of DVSA. Results showed that DVSA acted on the medial and lateral sensillum of larvae, resulting in the production of antifeedant signal in the medial and lateral sensillum, which possibly passed through the larval γ-aminobutyric acid system. Carboxylesterase and cytochrome P450 were the main targets of the decline of antifeedant effect in the later stage.

## Figures and Tables

**Figure 1 molecules-27-04464-f001:**
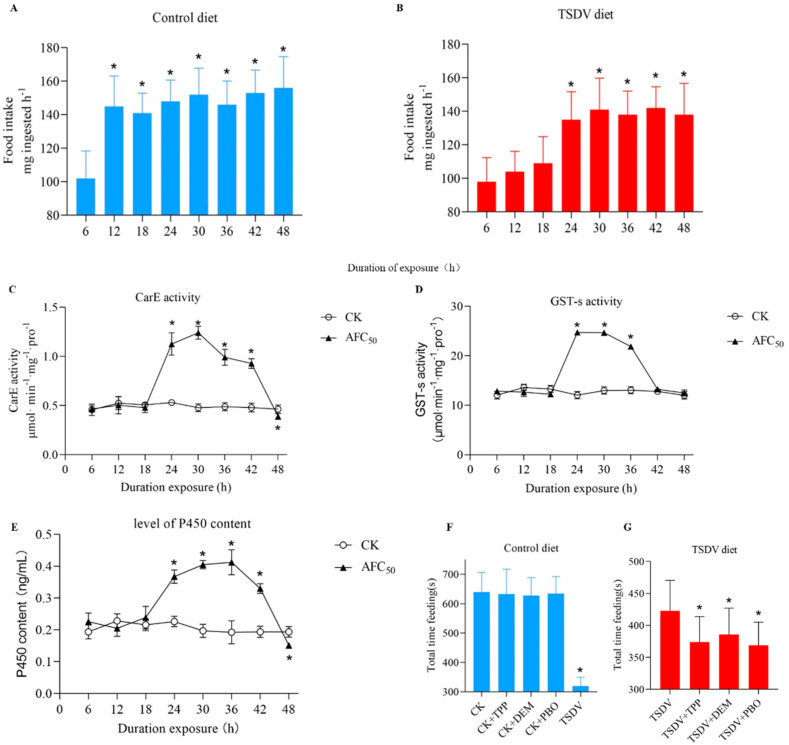
The effects of total saponins from *D. viscosa* on the detoxification enzyme in 4th-instar larva of *S. litura*. (**A**,**B**) The effect of consumption of a control diet or a AFC_50_ total saponins from *D. viscosa* diet by 4th-instar larvae of *S. litura* over a 48 h period. (**C**) The effects of total saponins from *D. viscosa* on the detoxification enzyme CarE activity in 4th-instar larva of *S. litura.* (**D**) The effects of total saponins from *D. viscosa* on the detoxification enzyme GST-s activity in 4th-instar larva of *S. litura*. (**E**) The effects of total saponins from *D. viscosa* on the detoxification enzyme P450 content in 4th-instar larva of *S. litura*. (**F**,**G**) Total saponins from *D. viscosa* affects detoxification system through CarE, GST-s, and P450. The effect of treatment with detoxification enzyme inhibitor triphenyl phosphate (TPP), diethyl maleate (DEM), and piperonyl butoxide (PBO) on total saponins from *D. viscosa* consumption time within 30 min. Data is presented as Mean ± SE (*n* = 10). ***** indicates significant differences according to Tukey post hoc test, *p* < 0.05.

**Figure 2 molecules-27-04464-f002:**
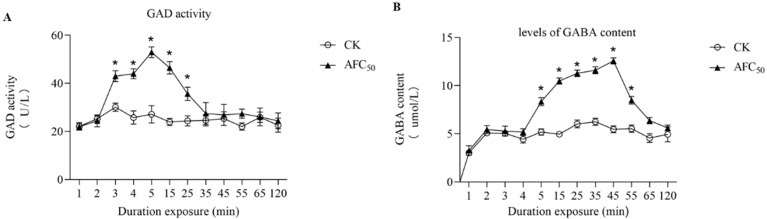
The effects of total saponins from *D. viscosa* on the GAD (**A**) activity and GABA (**B**) content in 4th-instar larva of *S. litura.* The GAD activity and GABA content are presented as Mean ± SE (*n* = 10). ***** indicates significant differences according to Tukey post hoc test, *p* < 0.05.

**Figure 3 molecules-27-04464-f003:**
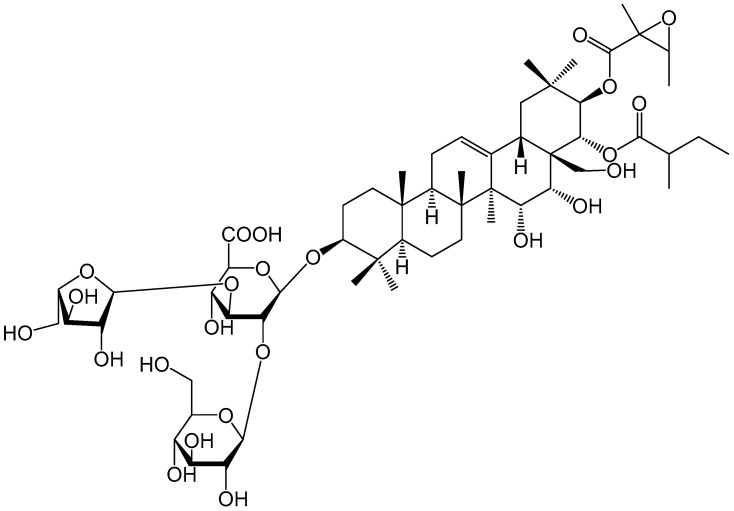
The chemical structure of *D. viscosa* saponin A.

**Figure 4 molecules-27-04464-f004:**
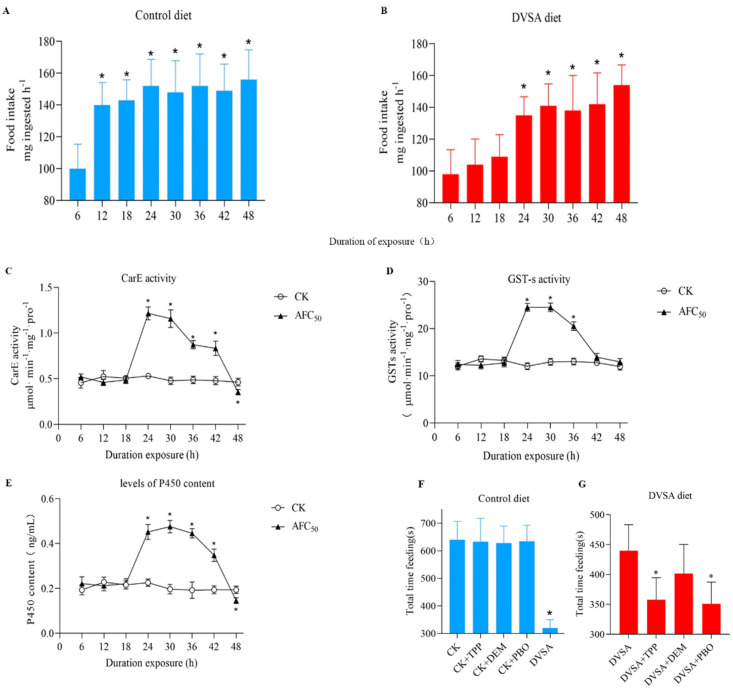
The effects of *D. viscosa* saponin A on the detoxification enzyme in 4th-instar larva of *S. litura.* (**A**,**B**) *D. viscosa* saponin A could change the feeding response of 4th-instar larvae of *S. litura.* The effect of consumption of a control diet or a AFC_50_
*D. viscosa* saponin A diet by 4th-instar larvae of *S. litura* over a 48 h period. (**C**) The effects of *D. viscosa* saponin A on the detoxification enzyme CarE activity in 4th-instar larva of *S. litura.* (**D**) The effects of *D. viscosa* saponin A on the detoxification enzyme GST-s activity in 4th-instar larva of *S. litura.* (**E**) The effects of *D. viscosa* saponin A on the detoxification enzyme P450 content in 4th-instar larva of *S. litura.* (**F**,**G**) *D. viscosa* saponin A affects detoxification system through CarE and P450. The effect of treatment with detoxification enzyme inhibitor triphenyl phosphate (TPP), diethyl maleate (DEM), and piperonyl butoxide (PBO), on *D. viscosa* saponin A consumption time within 30 min. Data is presented as Mean ± SE (*n* = 10). ***** indicates significant differences according to Tukey post hoc test, *p* < 0.05.

**Figure 5 molecules-27-04464-f005:**
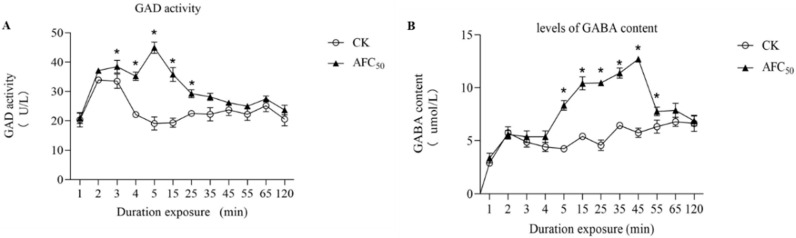
The effects of *D. viscosa* saponin A on the GAD activity (**A**) and GABA content (**B**) in 4th-instar larva of *S. litura.* The GAD activity and GABA content are presented as Mean ± SE (*n* = 10). ***** indicates significant differences according to Tukey post hoc test, *p* < 0.05.

**Figure 6 molecules-27-04464-f006:**
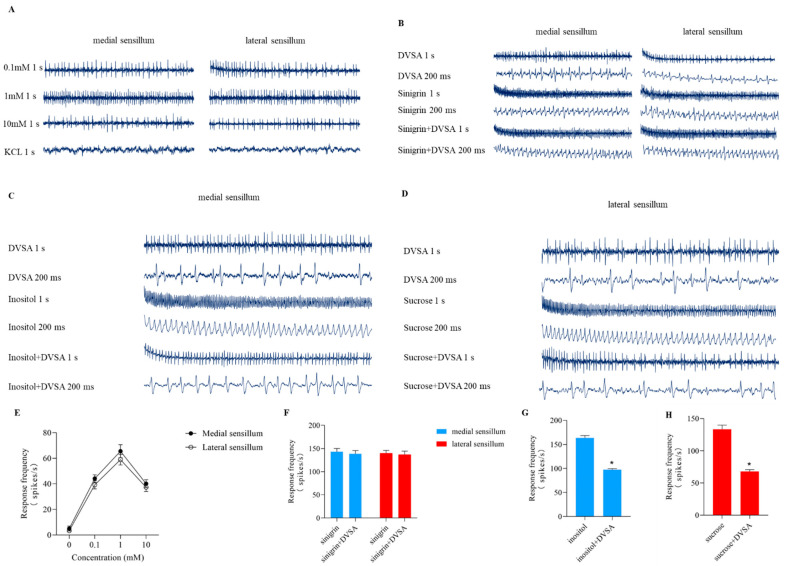
Effects of *D. viscosa* saponin A on the taste sensillum of *S*. *litura*. (**A**) Response (spikes/second) of lateral and medial styloconic sensillum of *S*. *litura* caterpillars to *D. viscosa* saponin A during the 1 s of stimulation. (**B**) Representative of the electrophysiological responses of medial styloconic sensillum to *D. viscosa* saponin A, sinigrin and their mixture in *S*. *litura*. (**C**) Representative of the electrophysiological responses of medial sensillum to *D. viscosa* saponin A, inositol, and their mixture in *S*. *litura*.(**D**) Representative of the electrophysiological responses of medial sensillum to *D. viscosa* saponin A, sucrose, and their mixture in *S*. *litura*. (**E**) Response frequency of the *D. viscosa* saponin A in medial sensillum and lateral sensillum. (**F**) Comparisons of response firing rate in 1 s to sinigrin and mixture with *D. viscosa* saponin A in the lateral styloconic sensillum. (**G**,**H**) Comparisons of response firing rate in 1 s to *D. viscosa* saponin A, inositol, sucrose, and their mixtures in the lateral styloconic sensillum and medial styloconic sensillum. Data is presented as Mean ± SE (*n* = 10). ***** indicates significant differences according to Tukey post hoc test, *p* < 0.05.

**Table 1 molecules-27-04464-t001:** Antifeedant activity of TSDV on 4th-instar larva of *Spodoptera litura*.

Concentration (μg/mL)	No. of Tested Insects	Feeding Area (mm^2^)	Inhibition Rate (%)
0	10	985.36 ± 86.79 ^a^	
500	10	687.68 ± 83.06 ^b^	30.21
1625	10	585.14 ± 45.89 ^c^	40.62
2750	10	450.36 ± 62.50 ^d^	54.33
3875	10	271.27 ± 38.30 ^e^	72.47
5000	10	149.38 ± 7.50 ^f^	84.84
Azadirachtin	10	302.70 ± 64.83 ^e^	69.26

The positive control was 500 μg/mL Azadirachtin (purity, ≥81%). Feeding area is presented as Mean ± SE (*n* = 10). Different superscript letters indicate significant differences according to Tukey post hoc test (*p* < 0.05).

**Table 2 molecules-27-04464-t002:** The effect of TSDV on developmental period on *Spodoptera litura*.

Concentration	Developmental Period/d				
	4th–6th Instar Larva	Female Pupae	Male Pupae	Female Adult	Male Adult
CK	12.10 ± 0.067 ^c,d^	12.73 ± 0.014 ^a^	11.11 ± 0.070 ^a^	6.84 ± 0.076 ^a^	3.88 ± 0.101 ^a^
AFC_10_	12.27 ± 0.094 ^b,c^	12.75 ± 0.063 ^a^	11.07 ± 0.047 ^a^	6.79 ± 0.083 ^a^	3.97 ± 0.056 ^a^
AFC_30_	12.42 ± 0.268 ^a,b^	12.83 ± 0.085 ^a^	11.07 ± 0.051 ^a^	6.78 ± 0.047 ^a^	3.64 ± 0.050 ^b^
AFC_50_	12.65 ± 0.106 ^a^	12.73 ± 0.065 ^a^	10.75 ± 0.084 ^b^	5.59 ± 0.076 ^b^	3.00 ± 0.035 ^c^

Developmental period is presented as Mean ± SE (*n* = 10). Different superscript letters indicate significant differences according to Tukey post hoc test, *p* < 0.05.

**Table 3 molecules-27-04464-t003:** The effect of TSDV on pupation rate and emergence rate in 4th-instar larva of *Spodoptera litura*.

Concentration	Pupation Rate	Emergence Rate
CK	100.00 ± 0.00 ^a^	88.00 ± 2.00 ^a^
AFC_10_	98.00 ± 2.00 ^a^	83.56 ± 2.65 ^a^
AFC_30_	84.00 ± 5.10 ^b^	69.12 ± 2.12 ^b^
AFC_50_	76.00 ± 5.10 ^b^	70.68 ± 2.76 ^b^

Pupation rate and emergence rate are presented as Mean ± SE (*n* = 10). Different superscript letters indicate significant differences according to Tukey post hoc test, *p* < 0.05.

**Table 4 molecules-27-04464-t004:** Antifeedant activity of DVSA on 4th-instar larva of *Spodoptera litura*.

Concentration (μg/mL)	No. of Tested Insects	Feeding Area (mm^2^)	Inhibition Rate (%)
0	10	985.36 ± 86.36 ^a^	
50	10	658.12 ± 89.57 ^b^	23.52
100	10	565.31 ± 45.93 ^c^	32.35
200	10	499.28 ± 55.33 ^c^	39.57
400	10	359.81 ± 39.11 ^d^	52.63
800	10	292.06 ± 20.95 ^d^	70.36
Azadirachtin	10	302.70 ± 64.83 ^d^	69.26

The positive control was 500 μg/mL Azadirachtin (purity, ≥81%). Feeding area is presented as Mean ± SE (*n* = 10). Different superscript letters indicate significant differences according to Tukey post hoc test, *p* < 0.05.

**Table 5 molecules-27-04464-t005:** The effect of DVSA on developmental period in 4th-instar larva of *Spodoptera litura*.

Concentration	Developmental Period/d				
	4th–6th Instar Larva	Female Pupae	Male Pupae	Female Adult	Male Adult
CK	12.09 ± 0.014 ^c^	12.68 ± 0.014 ^a^	11.11 ± 0.010 ^a^	6.84 ± 0.011 ^a^	3.88 ± 0.016 ^a^
AFC_10_	12.12 ± 0.012 ^c^	12.71 ± 0.015 ^a^	11.10 ± 0.012 ^a^	6.79 ± 0.013 ^a^	3.88 ± 0.011 ^a^
AFC_30_	12.48 ± 0.008 ^b^	12.70 ± 0.019 ^a^	10.92 ± 0.014 ^b^	6.08 ± 0.012 ^b^	3.90 ± 0.012 ^a^
AFC_50_	13.04 ± 0.011 ^a^	12.35 ± 0.018 ^b^	10.85 ± 0.016 ^c^	5.51 ± 0.017 ^c^	3.87 ± 0.016 ^a^

Developmental period is presented as Mean ± SE (*n* = 10). Different superscript letters indicate significant differences according to Tukey post hoc test, *p* < 0.05.

**Table 6 molecules-27-04464-t006:** The effect of DVSA on pupation rate and emergence rate in 4th-instar larva of *Spodoptera litura*.

Concentration	Pupation Rate	Emergence Rate
CK	100.00 ± 0.000 ^a^	96.00 ± 2.449 ^a^
AFC_10_	98.00 ± 2.000 ^a^	94.00 ± 4.000 ^a^
AFC_30_	90.00 ± 3.162 ^b^	79.89 ± 5.512 ^b^
AFC_50_	78.00 ± 3.741 ^c^	76.98 ± 2.403 ^b^

Pupation rate and emergence rate are presented as Mean ± SE (*n* = 10). Different superscript letters indicate significant differences according to Tukey post hoc test, *p* < 0.05.

## Data Availability

The data presented in this study are available on request from the corresponding author.

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
