# Peer review of "Antifeedant Mechanism of Dodonaea viscosa Saponin A Isolated from the Seeds of Dodonaea viscosa"

_molecules, 2022, doi:10.3390/molecules27144464_

Round 1
Reviewer 1 Report
This manuscript has the aim to evaluate the antifeedant mechanism of a specific saponin isolated from the seeds of Dodonaea viscosa, a shrub from the Sapindaceae family. The work seems seriously done, since many essays have been performed, statistical analysis is solid and the results are interesting. My only observations relate to the data presentation, that in some cases could deserve additional care. In the M&M section, it is not clear how many insects were used in each feeding essay, neither for detoxification enzymes activity (to term them “a group” is not enough). Table 6 reports that 10 insects were tested for each concentration, and the footnote to the same table refers to 20 replications; shall I assume that 10x20= 200 insects were used per each treatment? This point should be clarified in the M&M section.
Line 16 (Abstract): I do not get the meaning of this sentence. How this plant was used for conservation of water and soil?
Lines 83-84: check the opportunity to explain in the M&M section what AFC10, AFC30 etc. stand for.
Line 234 and 306: substitute “date” with “data”
Line 319: write “Pieris” with capital initial.
Line 321: do not start a sentence with “and”.
M&M line 352: add the classification (order and family) of the tested insects; furthermore, I am not sure that the terms “nymphs” (e.g., lines 84, 110, 114, and 119) and “larvae” (e.g., lines 86 and 93) can be used as synonyms.
Captions to figures 4 and 6: in my opinion, captions to figures should report only a brief description of the represented measurements, and longer statements (“We fed larvae…” and so on) should more properly find a place within the text.
Author Response
Dear editor,
Thank you very much for your kind decision on our manuscript and inviting us to a major revision of our manuscript. I would like to thank you and the two reviewers for taking precious time to review our manuscript and providing positive and insightful comments and very constructive suggestions.
We appreciate that all reviewers pointed out that this study evaluate the antifeedant mechanism of a specific saponin isolated from the seeds of Dodonaea viscosa. We have revised our manuscript accordingly and appropriate changes have been incorporated into our manuscript. Following is our response to the comments from reviewers.
Reviewer 1:
This manuscript has the aim to evaluate the antifeedant mechanism of a specific saponin isolated from the seeds of Dodonaea viscosa, a shrub from the Sapindaceae family. The work seems seriously done, since many essays have been performed, statistical analysis is solid and the results are interesting. My only observations relate to the data presentation, that in some cases could deserve additional care. In the M&M section, it is not clear how many insects were used in each feeding essay, neither for detoxification enzymes activity (to term them “a group” is not enough). Table 6 reports that 10 insects were tested for each concentration, and the footnote to the same table refers to 20 replications; shall I assume that 10x20= 200 insects were used per each treatment? This point should be clarified in the M&M section.
Response: Thank the reviewer for this good suggestion. We have added the insects quantity in each experiment in the M&M section. For Table 6, the experiment was carried out in triplicates, we have already changed the footnote to Table 6 accordingly. Sorry for the confusion.
- Line 16 (Abstract): I do not get the meaning of this sentence. How this plant was used for conservation of water and soil?
Response: Thank the reviewer for this suggestion. We have deleted this sentence.
- Lines 83-84: check the opportunity to explain in the M&M section what AFC10, AFC30 etc. stand for.
Response: Thank the reviewer for this suggestion. We have added the definition of AFC10, AFC30 as suggested in the M&M section.
- Line 234 and 306: substitute “date” with “data”
Response: Thank the reviewer for this suggestion. We have corrected the errors.
- Line 319: write “Pieris” with capital initial.
Response: Thank the reviewer for this suggestion. We have corrected the error.
- Line 321: do not start a sentence with “and”.
Response: Thank the reviewer for this suggestion. We have corrected this grammar mistake as suggested.
- M&M line 352: add the classification (order and family) of the tested insects; furthermore, I am not sure that the terms “nymphs” (e.g., lines 84, 110, 114, and 119) and “larvae” (e.g., lines 86 and 93) can be used as synonyms.
Response: Thank the reviewer for pointing this out. We have changed “nymphs” to “larvae”.
- Captions to figures 4 and 6: in my opinion, captions to figures should report only a brief description of the represented measurements, and longer statements (“We fed larvae…” and so on) should more properly find a place within the text.
Response: Thank the reviewer for this good suggestion. We changed the captions to figures with a brief description as the reviewer suggested.
Reviewer 2 Report
This manuscript provides interesting results and may be published in the journal after extensive revision. The authors report that saponin A extracted from Dodonaea viscosa showed adverse effects on two lepidopteran species, Spodoptera litura and Helicoverpa armigera. However, the experiments are not complete in that the growth performance was tested on both species while the experiment on the detoxification enzymes was examined on the former and electrode experiment on the latter. Although both moths are polyphagous, there are still remain issues that their studies did not assure that these species responded to the deterrents in the same way. In my opinion, the authors should concentrate the results of one species, especially S. litura. Other two major issues are:
1) dates and times of experiments should be referred to; 2) statistical tests should be explained for more in detail, especially on which tests were applied to which data to test what hypothesis. Also the authors referred to the statistics for each analysis in Results. Other minor comments are written below.
L31-33
The sentence is duplicated to LL20-22.
L33-34
It is still early to deduce that the potential of the extract from the plant can be used for biopesticide. No lethal effects of the extract were confirmed in the study. At least, the authors should test the reducible effects of the extract on the population of the pests.
Introduction
I suggest to describe the two lepidopteran species (or the insect more concentrated in the revised manuscript), especially on how much their damage is serious in crops, what managements are performed currently, what problems are brought in these managements and how the problems can be avoided or reduced by using biopesticides, etc.
L58-60
These insecticides are not banned in all countries.
L83
Show the statistics, such as the coefficient of determination, of the regression and make a significance test on it.
Table 1
Do not use “nymph” for metamorphotic species, for which “larva” must be used. As later mentioned, Duncan’s test must be avoided due to its higher likeliness of Type I error.
L176
What is 3 25 min?
L319
Capitalise “p” of pieris.
L359
Explain how the discs were prepared.
L384
Explain the kit. Is it a commercial product or self-made?
L398
How were the larvae maintained?
L444
Explain what tests were used which model(s).
L445
As previously mentioned, Duncan’s test should be avoided because of its higher likeliness of Type I error.
Author Response
Dear editor,
Thank you very much for your kind decision on our manuscript and inviting us to a major revision of our manuscript. I would like to thank you and the two reviewers for taking precious time to review our manuscript and providing positive and insightful comments and very constructive suggestions.
We appreciate that all reviewers pointed out that this study evaluate the antifeedant mechanism of a specific saponin isolated from the seeds of Dodonaea viscosa. We have revised our manuscript accordingly and appropriate changes have been incorporated into our manuscript. Following is our response to the comments from reviewers.
Reviewer 2:
This manuscript provides interesting results and may be published in the journal after extensive revision. The authors report that saponin A extracted from Dodonaea viscosa showed adverse effects on two lepidopteran species, Spodoptera litura and Helicoverpa armigera. However, the experiments are not complete in that the growth performance was tested on both species while the experiment on the detoxification enzymes was examined on the former and electrode experiment on the latter. Although both moths are polyphagous, there are still remain issues that their studies did not assure that these species responded to the deterrents in the same way. In my opinion, the authors should concentrate the results of one species, especially S. litura.
Response: Thank the reviewer for this good suggestion. We found that DVSA showed similar antifeedant activity in Spodoptera litura and Helicoverpa armigerat. Now we provide the antifeedant data of Helicoverpa armigerat in Figure 4A&C. Whether it is Spodoptera litura or Helicoverpa armigera, after taking DVSA for a few seconds, the tested larva stopped taking food, remained still for about 10 minutes. With the extension of time, the feeding amount gradually increased. The control group insects continued taking food for about 10 minutes. This result suggested that these species might respond to the deterrents in a similar way.
Other two major issues are:1) dates and times of experiments should be referred to;
Response: We have added the dates and times of experiments in the M&M section.
2) statistical tests should be explained for more in detail, especially on which tests were applied to which data to test what hypothesis. Also the authors referred to the statistics for each analysis in Results.
Response: As the reviewer suggested, we have added more detailed description of statistical analysis in Results and M&M section.
Other minor comments are written below.
- L31-33: The sentence is duplicated to LL20-22.
Response: Thank the reviewer for pointing this out. We have deleted the sentence at L31-33.
- L33-34: It is still early to deduce that the potential of the extract from the plant can be used for biopesticide. No lethal effects of the extract were confirmed in the study. At least, the authors should test the reducible effects of the extract on the population of the pests.
Response: Thank the reviewer for this suggestion. Currently we did not have evidence to support the lethal effects of the extract. We admit that it is too early to draw this conclusion. We already deleted this sentence.
- I suggest to describe the two lepidopteran species (or the insect more concentrated in the revised manuscript), especially on how much their damage is serious in crops, what managements are performed currently, what problems are brought in these managements and how the problems can be avoided or reduced by using biopesticides, etc.
Response: Thank the reviewer for this good suggestion. We have added the description of Spodoptera litura in the third paragraph in the Introduction.
- L58-60: These insecticides are not banned in all countries.
Response: Thank the reviewer for this suggestion. We have changed this sentence to “organochlorine pesticides are banned in China” as suggested.
- L83: Show the statistics, such as the coefficient of determination, of the regression and make a significance test on it.
Response: Thank the reviewer for this suggestion. We have added the coefficient of determination as suggested.
- Table 1: Do not use “nymph” for metamorphotic species, for which “larva” must be used. As later mentioned, Duncan’s test must be avoided due to its higher likeliness of Type I error.
Response: Thank the reviewer for this suggestion. We have corrected the errors. Tukey post hoc test was used instead of Duncan’s test in the revised manuscript.
- L176: What is 3 25 min?
Response: Thank the reviewer for pointing this out. It was a typo. The GAD activity in S. litura exposed to AFC50 TSDV diet increased significantly in 3-25 min. We have corrected the typo.
- L319: Capitalise “p” of pieris.
Response: We have done this.
- Explain how the discs were prepared.
Response: We have added the method of making leaf disc in the M&M section as suggested.
- L384: Explain the kit. Is it a commercial product or self-made?
Response: The kit was commercially available. The detailed information of the kit was provided.
- L398: How were the larvae maintained?
Response: We have provided the method of effects of DVSA on the taste sensillum of H. armigera in 4.8. Effects of DVSA on the taste sensillum of H. armigera in the M&M section.
- L444: Explain what tests were used which model(s).
Response: Thank the reviewer for this good suggestion. We have added the information about the tests used in each model.
- L445: As previously mentioned, Duncan’s test should be avoided because of its higher likeliness of Type I error.
Response: Thank the reviewer for this good suggestion. We have changed the Duncan’s test to Tukey post hoc test as suggested.
Round 2
Reviewer 2 Report
The most serious problem in this study is its design like ladder. Some experiments were carried out only for S. litura, one for H. armigera, and the others for both. Then the authors deduce the mechanisms of assumed antifeedant chemicals in the plant, and based on the physiological responses of the latter species, extend the impact of the substances extracted to all responses of both species to the feeding behaviour. As I pointed out in the previous review, the authors should test one species throughout the experiment.
The other serious problem is the preparation of the materials to test. The authors describe they repeated each experiment three times at L365-369, 381-385, 398-400, 411-413, 436-438. However, as long as the manuscript is read, the authors first prepared "one" sample from "one" plant, obtained 10 sub-samples from the prepared sample and then measured three times for each sub-sample. Thus, the replicates were obtained from only one sample. For replicates, different samples must have been prepared.
The objective of this study is rationale.
I strongly recommend the authors to carry out the experiment again, avoid the above problems.
Author Response
The most serious problem in this study is its design like ladder. Some experiments were carried out only for S. litura, one for H. armigera, and the others for both. Then the authors deduce the mechanisms of assumed antifeedant chemicals in the plant, and based on the physiological responses of the latter species, extend the impact of the substances extracted to all responses of both species to the feeding behaviour. As I pointed out in the previous review, the authors should test one species throughout the experiment.
Response: Thank you for the suggestion. As the reviewer suggested, we added the data of the effects of DVSA on the taste sensillum of S. litura. The results were shown in Figure 6. We also deleted all the data of H. armigera.
The other serious problem is the preparation of the materials to test. The authors describe they repeated each experiment three times at L365-369, 381-385, 398-400, 411-413, 436-438. However, as long as the manuscript is read, the authors first prepared "one" sample from "one" plant, obtained 10 sub-samples from the prepared sample and then measured three times for each sub-sample. Thus, the replicates were obtained from only one sample. For replicates, different samples must have been prepared.
Response: The replicates of our experiment were obtained from different samples. We corrected the errors in Materials and Methods.